# Scalable Vector Representation
# for Topological Data Analysis Based Classification

**Tananun Songdechakraiwut**                                      SONGDECHAKRA@WISC.EDU
*Department of Electrical and Computer Engineering, University of Wisconsin–Madison*

**Bryan M. Krause**
**Matthew I. Banks**
*Department of Anesthesiology & Department of Neuroscience, University of Wisconsin–Madison*

**Kirill V. Nourski**
*Department of Neurosurgery, University of Iowa*

**Barry D. Van Veen**
*Department of Electrical and Computer Engineering, University of Wisconsin–Madison*

**Editors:** Sophia Sanborn, Christian Shewmake, Simone Azeglio, Arianna Di Bernardo, Nina Miolane

## Abstract

Classification of large and dense networks based on topology is very difficult due to the computational challenges of extracting meaningful topological features from real-world networks. In this paper we present a computationally tractable approach to topological classification of networks by using principled theory from persistent homology and optimal transport to define a novel vector representation for topological features. The proposed vector space is based on the Wasserstein distance between persistence barcodes. The 1-skeleton of the network graph is employed to obtain 1-dimensional persistence barcodes that represent connected components and cycles. These barcodes and the corresponding Wasserstein distance can be computed very efficiently. The effectiveness of the proposed vector space is demonstrated using support vector machines to classify brain networks. This extended abstract is adapted from the extended work reported in Songdechakraiwut et al. (2022).

**Keywords:** Topological data analysis, classification, Wasserstein distance, networks, graphs

## 1. Introduction

Connected components and cycles are the most dominant and fundamental topological features of real networks. Many networks naturally organize into modules or connected components (Bullmore and Sporns, 2009; Honey et al., 2007). Similarly, cycle structure is ubiquitous and is often interpreted in terms of information propagation, redundancy and feedback loops (Kwon and Cho, 2007; Ozbudak et al., 2005; Weiner et al., 2002).

Here we present a novel *topological vector space* (TopVS) that embeds persistence barcodes for connected components and cycles. TopVS preserves the underlying distance in the original space of persistence barcodes while existing methods do not (Carrière and Bauer, 2019). The $p$-norm distance in TopVS is equivalent to the $p$-Wasserstein distance in the original barcode space. This equivalence allows the computation of summary statistics such as the mean of persistence barcodes to be easily performed in TopVS. The utility of TopVS is illustrated by classifying measured functional brain networks associated with different levels of arousal during administration of general anesthesia. TopVS performs very well compared to other topology-based approaches.

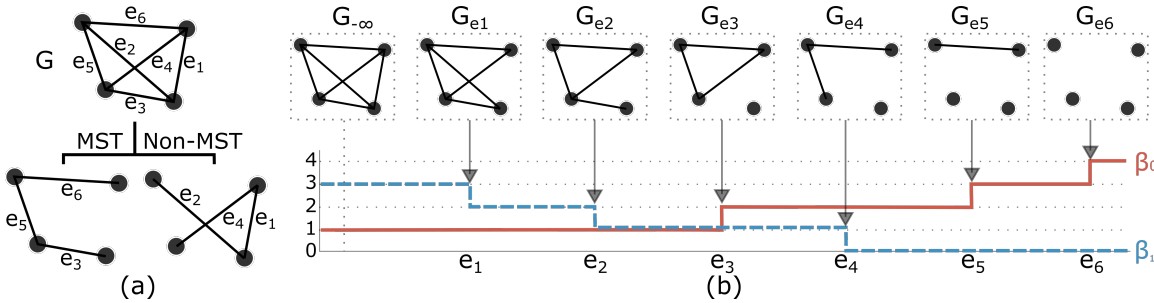

Figure 1: (a) Four-node network $G$ decomposes into its maximum spanning tree (MST) and a subnetwork with non-MST edge weights. (b) As the filtration value increases, the number of connected components $\beta_0$ monotonically increases while the number of cycles $\beta_1$ monotonically decreases.

## 2. Topological Space with Wasserstein Distance

**Graph Filtration** Define a network as an undirected weighted graph $G = (V, \boldsymbol{w})$ with a set of nodes $V$, and a weighted adjacency matrix $\boldsymbol{w} = (w_{ij})$. The number of nodes is denoted by $|V|$. Define a binary graph $G_\epsilon$ with the identical node set $V$ by thresholding the edge weights so that an edge between nodes $i$ and $j$ exists if $w_{ij} > \epsilon$. The binary graph is viewed as a simplicial complex consisting of only nodes and edges, that is, a 1-skeleton (Munkres, 2018). As $\epsilon$ increases, more and more edges are removed from the network $G$. Thus, we have a nested sequence of 1-skeletons: $G_{\epsilon_0} \supseteq G_{\epsilon_1} \supseteq \cdots \supseteq G_{\epsilon_k}$, where $\epsilon_0 \leq \epsilon_1 \leq \cdots \leq \epsilon_k$ are called filtration values. This sequence of 1-skeletons is called a *graph filtration*.

**One Dimensional Persistence Barcodes** Persistent homology keeps track of the birth and death of topological features over filtration values $\epsilon$. A topological feature that is born at a filtration $b_i$ and persists up to a filtration $d_i$, is represented as a 2-dimensional point $(b_i, d_i)$ in a plane. A set of all the points $\{(b_i, d_i)\}$ is called *persistence barcode* (Ghrist, 2008). In the 1-skeleton, the only non-trivial topological features are connected components (0-dimensional topological features) and cycles (1-dimensional topological features). As $\epsilon$ increases, the number of connected components $\beta_0(G_\epsilon)$ and cycles $\beta_1(G_\epsilon)$ are monotonically increasing and decreasing, respectively (Songdechakraiwut et al., 2021). Thus, the representation of the connected components can be simplified to a collection of sorted birth values $B(G) = \{b_i\}_{i=1}^{|V|-1}$. Similarly, we can simplify the representation of the cycles as a collection of sorted death values $D(G) = \{d_i\}$. The example network of Figure 1 has $B(G) = \{e_3, e_5, e_6\}$ and $D(G) = \{e_1, e_2, e_4\}$.

**Wasserstein Distance Simplification** The Wasserstein distance between the 1-dimensional barcodes of the graph filtration can be obtained using a closed-form solution. Let $G_i$ be a network. Its underlying probability density function on the persistence barcodes for connected components is defined in the form of Dirac masses (Turner et al., 2014) as

$$f_{G_i, B}(x) := \frac{1}{|B(G_i)|} \sum_{b \in B(G_i)} \delta(x - b)$$

where $\delta(x - b)$ is a Dirac delta centered at the point $b$. Then the empirical distribution is the integration of $f_{G_i,B}$ as

$$F_{G_i,B}(x) = \frac{1}{|B(G_i)|} \sum_{b \in B(G_i)} \mathbb{1}_{b \leq x}$$

where $\mathbb{1}_{b \leq x}$ is an indicator function taking the value 1 if $b \leq x$, and 0 otherwise. A pseudoinverse of $F_{G_i,B}$ is defined as $F_{G_i,B}^{-1}(z) = \inf\{b \in \mathbb{R} \,|\, F_{G_i,B}(b) \geq z\}$, i.e., $F_{G_i,B}^{-1}(z)$ is the smallest $b$ for which $F_{G_i,B}(b) \geq z$. Then the empirical Wasserstein distance for connected components has a closed-form solution in terms of pseudoinverses as

$$W_{p,B}(G_1, G_2) = \left( \int_0^1 |F_{G_1,B}^{-1}(z) - F_{G_2,B}^{-1}(z)|^p \, dz \right)^{1/p}.$$

Similarly, the Wasserstein distance for cycles $W_{p,D}(G_1, G_2)$ is defined in terms of empirical distributions for death sets $D(G_1)$ and $D(G_2)$.

The empirical Wasserstein distances $W_{p,B}$ and $W_{p,D}$ are approximated by computing the Lebesgue integration numerically as follows. Let $\widehat{B}(G_1) = \{F_{G_1,B}^{-1}(1/m), ..., F_{G_1,B}^{-1}(m/m)\}$ and $\widehat{D}(G_1) = \{F_{G_1,D}^{-1}(1/n), ..., F_{G_1,D}^{-1}(n/n)\}$ be pseudoinverses of network $G_1$ sampled with partitions of equal intervals. Let $\widehat{B}(G_2)$ and $\widehat{D}(G_2)$ be sampled pseudoinverses of network $G_2$ with the same partitions of $m$ and $n$, respectively. Then the approximated Wasserstein distances are given by $\widehat{W}_{p,B}(G_1, G_2) = \left( \frac{1}{m^p} \sum_{k=1}^m \left| F_{G_1,B}^{-1}(k/m) - F_{G_2,B}^{-1}(k/m) \right|^p \right)^{1/p}$ and $\widehat{W}_{p,D}(G_1, G_2) = \left( \frac{1}{n^p} \sum_{k=1}^n \left| F_{G_1,D}^{-1}(k/n) - F_{G_2,D}^{-1}(k/n) \right|^p \right)^{1/p}$.

**Vector Representation of Persistence Barcodes**  A collection of 1-dimensional persistence barcodes together with the Wasserstein distance is a metric space. 1-dimensional persistence barcodes can be embedded into a vector space that preserves the Wasserstein metric on the original space of persistence barcodes as follows. Let $G_1, G_2, ..., G_N$ be $N$ observed networks possibly with different node sizes. Let $F_{G_i,B}^{-1}$ be a pseudoinverse of network $G_i$. The vector representation of a persistence barcode for connected components in network $G_i$ is defined as a vector of the pseudoinverse sampled at $1/m, 2/m, ..., m/m$. That is, $\boldsymbol{v}_{B,i} := \left( F_{G_i,B}^{-1}(1/m), F_{G_i,B}^{-1}(2/m), ..., F_{G_i,B}^{-1}(m/m) \right)^\top$. A collection of these vectors $M_B = \{\boldsymbol{v}_{B,i}\}_{i=1}^N$ with the $p$-norm $|| \cdot ||_p$ induces the $p$-norm metric $d_{p,B}$ given by $d_{p,B}(\boldsymbol{v}_{B,i}, \boldsymbol{v}_{B,j}) = ||\boldsymbol{v}_{B,i} - \boldsymbol{v}_{B,j}||_p = m\widehat{W}_{p,B}$. Thus, for $p = 1$ the proposed vector space describes Manhattan distance, $p = 2$ Euclidean distance, and $p \to \infty$ the maximum metric, which in turn correspond to the earth mover's distance ($W_1$) (Rubner et al., 2000), 2-Wasserstein distance ($W_2$), and the bottleneck distance ($W_\infty$) (Kerber et al., 2017), respectively, in the original space of persistence barcodes. Similarly, we can define a vector space of persistence barcodes for cycles $M_D = \{\boldsymbol{v}_{D,i}\}_{i=1}^N$ with the $p$-norm metric $d_{p,D}$. The normed vector space $(M_B, d_{p,B})$ describes topological space of connected components in networks, while $(M_D, d_{p,D})$ describes topological space of cycles in networks.

The topology of a network viewed as a 1-skeleton is *completely* characterized by connected components and cycles. Thus, we can fully describe the network topology using both $M_B$ and $M_D$ as follows. Let $M_B \times M_D = \{(\boldsymbol{v}_{B,i}, \boldsymbol{v}_{D,i}) \,|\, \boldsymbol{v}_{B,i} \in M_B, \boldsymbol{v}_{D,i} \in M_D\}$ be

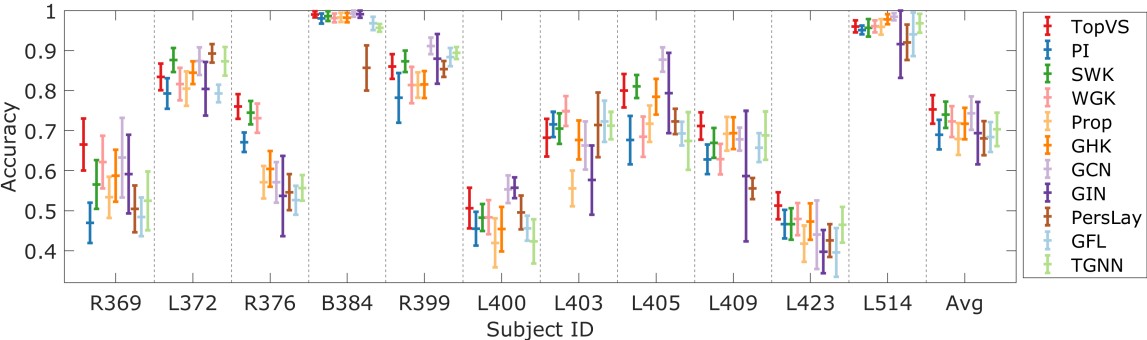

Figure 2: Accuracy classifying brain networks within individual subjects. The last column displays the average accuracy obtained across all subjects. The center markers and bars depict the means and standard deviations obtained over 100 trials.

the Cartesian product between $M_B$ and $M_D$ so the vectors in $M_B \times M_D$ are the concatenations of $\boldsymbol{v}_{B,i}$ and $\boldsymbol{v}_{D,i}$. For this product space to represent meaningful topology of network $G_i$, the vectors $\boldsymbol{v}_{B,i}$ and $\boldsymbol{v}_{D,i}$ must be a network decomposition, as illustrated in Figure 1. Thus $\boldsymbol{v}_{B,i}$ and $\boldsymbol{v}_{D,i}$ are constructed by sampling their psudoinverses with $m = \mathcal{V} - 1$ and $n = 1 + \frac{\mathcal{V}(\mathcal{V}-3)}{2}$, respectively, where $\mathcal{V}$ is a free parameter indicating a reference network size. The metrics $d_{p,B}$ and $d_{p,D}$ can be put together to form a $p$-product metric $d_{p,\times}$ on $M_B \times M_D$ as

$$d_{p,\times}\big((\boldsymbol{v}_{B,i}, \boldsymbol{v}_{D,i}), (\boldsymbol{v}_{B,j}, \boldsymbol{v}_{D,j})\big) = \big([d_{p,B}(\boldsymbol{v}_{B,i}, \boldsymbol{v}_{B,j})]^p + [d_{p,D}(\boldsymbol{v}_{D,i}, \boldsymbol{v}_{D,j})]^p\big)^{1/p}$$
$$= \big([m\widehat{W}_{p,B}]^p + [n\widehat{W}_{p,D}]^p\big)^{1/p},$$

where $(\boldsymbol{v}_{B,i}, \boldsymbol{v}_{D,i}), (\boldsymbol{v}_{B,j}, \boldsymbol{v}_{D,j}) \in M_B \times M_D$, $m = \mathcal{V} - 1$ and $n = 1 + \frac{\mathcal{V}(\mathcal{V}-3)}{2}$. Thus, $d_{p,\times}$ is a weighted combination of $p$-Wasserstein distances, and is simply the $p$-norm metric between vectors constructed by concatenating $\boldsymbol{v}_{B,i}$ and $\boldsymbol{v}_{D,i}$. The normed vector space $(M_B \times M_D, d_{p,\times})$ is termed *topological vector space* (TopVS). A direct consequence of the equality is that the mean of persistence barcodes under the approximated Wasserstein distance is equivalent to the sample mean vector in TopVS. In addition, the proposed vector representation is highly interpretable because persistence barcodes can be easily reconstructed from vectors by separating sorted births and deaths.

## 3. Application to Functional Brain Networks

**Dataset**    We evaluate our method using a brain network dataset from the anesthesia study reported by Banks et al. (2020) (see Appendix A for details). The measured brain networks are based on eleven neurosurgical patients during administration of increasing doses of the general anesthetic propofol prior to surgery. Each segment is labeled as one of the three arousal states: pre-drug *wake*, *sedated* but responsive to command, or *unresponsive*.

**Classification performance evaluation**    We compare the classification performance of the proposed TopVS relative to that of several state-of-the-art methods. While nearly any classifier may be used with TopVS, here we illustrate results using the $C$-support vector

machine (SVM) (Chang and Lin, 2011) with the linear kernel, which maximizes Wasserstein distance-based margin. The performance of TopVS is compared to ten other methods for persistence barcodes, graph kernels and graph neural networks including persistence image (PI) (Adams et al., 2017), sliced Wasserstein kernel (SWK) (Carriere et al., 2017), persistence weighted gaussian kernel (PWGK) (Kusano et al., 2016), propagation kernel (Prop) (Neumann et al., 2016), graph hopper kernel (GHK) (Feragen et al., 2013), graph convolutional networks (GCN) (Kipf and Welling, 2017), graph isomorphism network (GIN) (Xu et al., 2019), PersLay (Carrière et al., 2020), graph filtration learning (GFL) (Hofer et al., 2020) and topological graph neural network (TGNN) (Horn et al., 2022). We apply a nested CV comprising an outer loop of stratified 2-fold CV and an inner loop of stratified 3-fold CV, for each subject. Since we may get a different split of data folds each time, we perform the nested CV for 100 trials and report an average accuracy score and standard deviation for each subject. We also average these individual accuracy scores across subjects ($11 \times 100$ scores) to obtain an overall accuracy.

**Results** Figure 2 compares classification accuracy for individual subjects. There is variability in performance across subjects and across methods. In most subjects all methods perform relatively well. The consistently poorer performance of PI, Prop, GIN, PersLay and GFL is evident in the lower overall performance. TopVS is demonstrated to perform favorably against the graph neural network classification methods. The results suggest that the use of complex classification methods, such as GCN, GIN, PersLay, GFL and TGNN, does not result in significant increase in generalizability when classifying brain networks. Our TopVS method is consistently among the best performing classifiers, resulting in the higher overall performance.

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

## Appendix A. Brain Network Dataset

Brain network data were obtained from eleven neurosurgical patients between 19 and 59 years old as described in Table 1. The patients were undergoing chronic invasive intracranial electroencephalography (iEEG) monitoring as part of their treatment for medically refractory epilepsy. The Code of Ethics of the World Medical Association (Declaration of Helsinki) for experiments involving humans was followed for all the experiments. The University of Iowa Institutional Review Board and the National Institutes of Health approved all research protocols, and written informed consent was obtained from all subjects. Acquisition of clinically required data was not impeded by the research and subjects were free to rescind their consent whenever they wished without interfering with their clinical evaluation. Subdural and depth electrodes (Ad-Tech Medical, Oak Creek, WI) used to obtain all research data were located by the team of epileptologists and neurosurgeons based solely on needs for clinical evaluation of the patients. Data collected in the operating room prior to electrode removal, before and during induction of general anesthesia with propofol were used to create the brain network dataset. Full description of the method for obtaining the brain network dataset and experimental procedure is provided in (Banks et al., 2020).

Table 1: Brain network dataset.

| Subject | Age | Gender | Network size |
|---------|-----|--------|--------------|
| R369 | 30 | M | 199 |
| L372 | 34 | M | 174 |
| R376 | 48 | F | 189 |
| B384 | 38 | M | 89 |
| R399 | 22 | F | 175 |
| L400 | 59 | F | 126 |
| L403 | 56 | F | 194 |
| L405 | 19 | M | 127 |
| L409 | 31 | F | 160 |
| L423 | 51 | M | 152 |
| L514 | 46 | M | 118 |

