# OpenReview forum: "Scalable Vector Representation for Topological Data Analysis Based Classification"
_NeurIPS.cc/2022/Workshop/NeurReps — NeurReps 2022 Poster_

### Official Review · Reviewer_HSSP · 2022-10-14
**Topological classification in a Wasserstein distance based vector space**

**Confidence:** 4
**Soundness:** 3
**Presentation:** 3
**Contribution:** 2
**Overall Rating:** 5

**Summary:**

The manuscript proposes an approach to the classification of weighted graphs $G$ based on topological feature extraction. To this end a filtration of the graph is constructed using the edge-weights as a filtration parameter. A superlevel set filtration is considered, meaning that an edge $e_{ij}$ is included at scale \epsilon if and only if the edge weight $w_{ij}$ exceeds $\epsilon$, and $\epsilon$ is increasing. So at the smallest edge weight the graph is fully connected, i.e. it consists of one connected component and all possible 1-dimensional loops in $G$. Increasing the filtration parameter has the effect that loops will die (number of loops is monotonically decreasing) and connected components are born with the number of connected components monotonically increasing. The collection $D_G$ of death-times of the loops, and the collection $B_G$ of birth times of connected components together describe the topology of the graphs. A certain distance between these features collected from different graphs then serve as proxies for distances between the graphs. This is being described below.

It is also observed that since each death time $d_i$ of a loop and birth time $b_j$ of a connected components correspond to an edge (with the corresponding edge weight being $d_i$ or $b_j$, respectively), the two sets $D_G$ and $B_G$, split the totally connected graph into two subgraphs, the minimum spanning tree of the graph and its complement. This means that in a fully connected graph with $V$ vertices, there are $m = V-1$ birth times, and $n = \binom{V}{2} - (V- 1) = 1 + \frac{V(V-3)}{2}$ many loops.

Identifying the sets $D_G$ and $B_G$, with one-dimensional (discrete) empirical measures $F_{D_G}$ and $F_{B_G}$, respectively, and given two graphs $G_1,G_2$, say, the $p$-Wasserstein distances $W_p(F_{D_1}, F_{D_2})$ and $W_p(F_{B_1}, F_{B_2})$ are the being used (where here $F_{D_1}$ is short for $F_{D_{G_1}}$, etc.). It is a well-known fact that these $p$-Wasserstein distances can be written as $W^p_p(F_{D_1},F_{D_2}) = \int_0^1 (F_{D_1}^{-1}(t) - F^{-1}_{D_2}(t))^pdt$ (and similarly for $D$ replaced by $B$). Since these distributions are discrete, this integral becomes the weighted sum, or  $ \| \tilde D_1 - \tilde D_2\|_p^p/n$, where the $n$-dimensional vectors $\tilde D_1, \tilde D_2$ are the sorted death values, i.e. the so-called order statistics of the death times (or the vector of quantiles).

A similar expression holds for the Wasserstein distance between the birth times. In other words, the $p$-Wasserstein distance between the discrete distributions $F_{D_1}$ and $F_{D_2}$ is equal to the distance in $p$-norm between the corresponding quantiles up the a factor of $1/n$ (and $1/m$ in case of the birth times).

Considering the direct product of the order statistics, gives a distance

$$d_{p,\times}((\tilde B_{G_1},\tilde B_{G_2}), (\tilde D_{G_1},\tilde D_{G_2})) = (\| \tilde D_1 - \tilde D_2\|_p + \| \tilde B_1 - \tilde B_2\|_p)^{1/p},$$

which can also be expressed as

$$d_{p,\times}((\tilde B_{G_1},\tilde B_{G_2}), (\tilde D_{G_1},\tilde D_{G_2})) = (n W^p_p(F_{D_1},F_{D_2}) + m W^p_p(F_{B_1},F_{B_2}))^{1/p},$$

which is a weighted combination of Wasserstein distances between the distributions.

In other words, for any graph $G$, the concatenated vector of quantiles of births $B_G$ and deaths $D_G$ serves as a topological feature vector of the graph, and the space of all such vectors taken together equipped with the above norm, is termed topological vector space (TopVec).

Given a set of networks or graphs, these pairwise distances between the graphs can now being used for classification. The performance of this classification is illustrated on a brain network by comparing it to various other classification methods using a cross-validation approach. Classification based on TopVec is shown to be competitive.



**Questions:**

1) The approach is formulated in such a way that all the networks considered have to have the same number of vertices. I am wondering whether this can be relaxed by considering refinements of the number of quantiles being used.

2) The classification procedure in described in section 3 is missing some details. Maybe I missed it, but I did not find a description of how exactly classification is performed based on the pairwise distances that are constructed. This information is also missing for other graph representations that are being used in the comparisons. (The Appendix mentions C-SVMs...) Also, nothing is being said about which value of $p$ is chosen. I think it would be helpful for the reader if this could be added.

3) If I am not mistaken, then the relation between the $p$-distances of the quantile vectors and the Wasserstein distances have a typo. Using the notation of the manuscript, believe it should be $m W^p_{p,B}$ rather than $(mW_{p,B})^p$ (and similarly for $B$ replaced by $D$) - see page 3, displayed formula towards the bottom of the page).

3) Page 2, paragraph entitled "Vector Representation of Persistence Barcodes", line 5: There, the quantile function is termed the "pseudo-inverse of the network", which to me did not make much sense.

4) Page 2: paragraph entitled "Wasserstein Distance Simplification":
- There, a sum of Dirac measures is termed a probability density. In my mind this is not correct, and it should rather be an (empirical) distribution.
- Also, the "integration of the density" presented in the same paragraph is termed the empirical distribution. To me this is the empirical distribution function (edf), rather than the distribution itself (which is the sum of Dirac measures).
- Constants $m$ and $n$ are being used in the construction of the sets $\widehat B(G)$ and $\widehat D(G)$, respectively. Later, $m$ and $n$ are specified, but in this paragraph nothing is said about how to choose $m$ and $n$, which was confusing at the first reading.


**Limitations:**

I did not find a discussion on the limitations. (I did mention one above, regarding the number of vertices of the networks having to be equal.)

**Recommended Decision:**

2: Borderline

**Relevance:**

3: Solid fit

**Strengths And Weaknesses:**

Strength:

1) The manuscript proposes a classification approach based on topological feature extraction that is well motivated and fits to the topic of the workshop.

2) The proposed method is computationally feasible, and competitive.

3) Overall, the presentation of the material is quite accessible.

4) The list of references is quite extensive.

Weaknesses:

1) The idea of using the distribution of birth and death times $F_{B_1}$ and $F_{D_1}$ of connected components and 1-dimensional loops in a graph $G$, respectively, as topological features for classification has already been proposed in Songdechakraiwut et al. (2021). There, an unweighted combination of $2$-Wasserstein distances between the distribution of birth times and the distribution of death times, respectively, has been proposed, and it also has been shown that the $2$-Wasserstein distance essentially is the distance in $2$-norm of the respective vectors of order statistics. Also the mentioned facts about the sets $D_G$ and $B_G$ splitting the graph into two disjoint subgraphs, etc. are presented in this paper. Even though Songdechakraiwut et al. (2021) use this observation in a different context (graph matching), to me this somewhat weakens the originality of the submitted manuscript.

2) The performance of the proposed method is illustrated by an application to brain networks. While this shows some good performance of the classification methodology, there is no novel scientific insight that is obtained from this.

3) As far as I could see, there is no novel mathematical contribution.


**Submission Track:**

Extended Abstract (4 Page)

---

### Official Review · Reviewer_fKzz · 2022-10-16
**Similar results have been presented before**

**Confidence:** 4
**Soundness:** 3
**Presentation:** 3
**Contribution:** 2
**Overall Rating:** 6

**Summary:**

This paper presents a computationally tractable approach to topological classification of networks by using principled theory from persistent homology and optimal transport to define a novel vector representation for topological features. The proposed vector space is based on the Wasserstein distance between persistence barcodes. The 1-skeleton of the network graph is employed to obtain 1-dimensional persistence barcodes that represent connected components and cycles. These barcodes and the corresponding Wasserstein distance can be computed very efficiently. The effectiveness of the proposed vector space is demonstrated using support vector machines to classify brain networks.

**Questions:**

The only issue I am concerned is that similar results have been presented in other venues.

**Recommended Decision:**

2: Borderline

**Relevance:**

3: Solid fit

**Strengths And Weaknesses:**

Overall, this paper is well-written and presents interesting and relevant topic. Also experimental results show the effectiveness of the proposed method.

**Submission Track:**

Extended Abstract (4 Page)

---

### Official Review · Reviewer_Zy84 · 2022-10-18
**This paper presents a new graph representation with topological properties and evaluates it on a real-world dataset.**

**Confidence:** 3
**Soundness:** 2
**Presentation:** 3
**Contribution:** 2
**Overall Rating:** 5

**Summary:**

This paper devises a vector representation for weighted, undirected graphs that is based on topological information derived from a persistent homology filtration procedure. For two given graphs, the p-norms between two representation vectors in representation vector space corresponds to the approximate p-Wasserstein distances between the persistence barcodes of the two graphs. The so-defined vector representation is applied in a classification task on graphs from a real-world neuroscience dataset.


[C1] Definition of a vector representation for graphs, which is based on a filtration procedure such that the p-norms between vectors correspond to the (approximate) p-Wasserstein distances between the persistence barcodes of two graphs.
[C2] Application of the proposed representation to a real-world classification task and benchmarking against standard graph embeddings.


**Questions:**

On page 2: [ … is represented as a 2-dimensional point ($b_i, d_i$) in a plane.]
I was slightly confused by the wording. I suggest changing this to “... is represented by a point ($b_i, d_i$) in a 2-dimensional plane”.


On page 2: Figure 1:
I would suggest clarifying which definition of “cycles” the authors use in their paper. With – what I understand to be the standard definition of a cycle – I find more than 3 cycles in the graph G. Examples for distinct cycles are (Numbering the nodes from leftmost to rightmost):
(123), (234), (341), (412), (1234)

In such a case, the figure 1 (b) would need to be corrected. At filtration step $e_1$ the graph then would lose more than just one cycle.


In the supplementary on page 12: [... The removal of a edge must result in either the birth of a connected component or the death of a cycle.]
Depending on the definition of cycles (see previous question), this statement should be adapted to: *... must result in either the birth of a connected component or the death of one or multiple cycles.*. An example of a death of multiple cycles would be given by Figure 1(b) at filtration $e_1$.
If this is the definition the authors use, the next sentence should also be corrected: The number of cycles $G_{-\infty}$ could be larger than the proposed value (c.f. discussion above)


**Limitations:**

To my understanding, the authors have not elaborated on the limitations of the given representation in particular. It would be good to see a discussion of:
- Relevance of the fixed graph size for comparable representations: Do the authors foresee a way to also compare graphs of different sizes?
- Further limitations? Next steps?


**Recommended Decision:**

2: Borderline

**Relevance:**

3: Solid fit

**Strengths And Weaknesses:**

*Clarity*:
The setup of edge weight filtration is nicely explained and illustrated by Fig 1. And the ordering of births and deaths of connected components and cycles respectively is theoretically sound.
The definition of “cycle” in the paper seems to require further explanation (see 2nd question) or might hint at a slight error in the paper (more than one cycle can die at a time), which would however not affect the main result. I hope the authors will clarify this part in the questions section.

*Quality*:
While there might be a potential mistakes in counting cycles (pending definition of “cycles” for the paper), the main result of the paper – the definition of the vector representation – is still valid, and so is the connection to the Wasserstein distance.

Some claims (e.g. “The topology of a network viewed as 1-skeleton is completely characterized by connected components and cycles”) should be supported, and I recommend the authors to cite a relevant source or provide a proof in the appendix.

It would be great to have a more detailed discussion of the merits and limitations of the proposed vector representation, especially given that the fairly large error bars on the chosen real-world dataset that overlap with many competitor models, most notably GCN. To this effect, a comparison a few other (benchmark) datasets / synthetic datasets might also be helpful to strengthen the author’s message and more clearly assess the usefulness of the proposed representation.

*Originality*:
To my understanding, one of the key insights in the paper is the ordering of birth and death values, which enables the definition of vector representations for fixed size graphs. This has been noted before in the cited reference (Songdechakraiwut et al. 2021), but has to my understanding not been used to create a graph representation. In this sense, the work is original. I cannot comment on whether this vector representation is the first to respect topological information, as I am not familiar enough with the topical literature.


**Submission Track:**

Extended Abstract (4 Page)

---

### Decision · Program_Chairs · 2022-10-21

Accept (Poster)